# A Q-Learning Based Scheme for Neighbor Discovery and Power Control in Marine Opportunistic Networks [note 1]

**DOI:** 10.3390/s25185720

**Published:** 2025-09-13

**Authors:** Jiahui Zhang, Shengming Jiang, Jinyu Duan

**Affiliations:** College of Information Engineering, Shanghai Maritime University, Shanghai 201306, China; smjiang@shmtu.edu.cn (S.J.); 202340310005@stu.shmtu.edu.cn (J.D.)

**Keywords:** neighbor discovery, opportunistic network, MAC layer, Q-learning, power control

## Abstract

Opportunistic networks, as an emerging ad hoc networking technology, the sparse distribution of nodes poses significant challenges to data transmission. Additionally, unlike static nodes in traditional ad hoc networks that can replenish energy on demand, the inherent mobility of nodes further complicates energy management. Thus, selecting an energy-efficient neighbor discovery algorithm is critical. Passive listening conserves energy by continuously monitoring channel activity, but it fails to detect inactive neighboring nodes. Conversely, active probing discovers neighbors by broadcasting probe packets, which increases energy consumption and may lead to network congestion due to excessive probe traffic. As the primary communication nodes in the maritime environment, vessels exhibit high mobility, and networks in oceanic regions often operate as opportunistic networks. To address the challenge of limited energy in maritime opportunistic networks, this paper proposes a hybrid neighbor discovery method that combines both passive and active discovery mechanisms. The method optimizes passive listening duration and employs Q-learning for adaptive power control. Furthermore, a more suitable wireless communication model has been adopted. Simulation results demonstrate its effectiveness in enhancing neighbor discovery performance. Notably, the proposed scheme improves network throughput while achieving up to 29% energy savings at most during neighbor discovery.

## 1. Introduction

With the continuous development of communication technologies, opportunistic networks, an emerging technology stemming from wireless mobile ad hoc networks, have gradually found widespread applications in various fields [1,2]. Opportunistic networks are a type of mobile ad hoc network that do not require a complete path between the source node and the destination node. Instead, they facilitate information transfer through a simple and reliable data forwarding mechanism of “store-carry-forward”. Neighbor discovery, as the most fundamental aspect in opportunistic networks, provides necessary information for subsequent network operations [3,4]. Only by exhaustively detecting nodes in the network can we obtain the network’s topology, subsequently identify routes, determine routing algorithms, establish minimum spanning trees based on neighbor information for more efficient propagation, and so forth [5].

Due to their mobility, nodes in opportunistic networks cannot replenish energy as readily as nodes in traditional ad hoc networks. Data transmission in opportunistic networks also heavily relies on neighbor discovery. If a node fails to accurately discover its neighbors, it may end up waiting for forwarding opportunities, which increases forwarding delays, leads to data transmission failures, and results in extreme waste of node energy. The proposal of maritime internet networks aims to meet the diverse needs of marine users across aerial, surface, and underwater environments. It integrates various opportunistic network systems, including shore-based wireless networks, wireless ad-hoc networks, unmanned aerial vehicle (UAV) opportunistic networks, high-altitude platforms, and satellite communications. Due to their inherently complex and dynamic nature, these networks exhibit enhanced adaptability. However, their opportunistic forwarding mechanisms also introduce additional uncertainty. To address neighbor discovery tasks in maritime internet networks, it is critically important to develop an energy-efficient and high-performance neighbor discovery scheme. This paper proposes an energy-efficient neighbor discovery method for maritime opportunistic networks, which integrates active-passive listening with Q-learning-based power control. By dynamically adjusting both passive listening duration and transmission power, the approach significantly reduces energy consumption in mobile opportunistic and wireless ad-hoc networks.

The method offers a novel solution for neighbor discovery and network formation in maritime internet environments through the joint optimization of passive listening intervals and active probing power. It introduces Q-learning into highly dynamic, delay-tolerant, and severely resource-constrained oceanic opportunistic networks, thereby avoiding blind energy expenditure. Moreover, this strategy extends the operational lifetime of unattended marine network nodes that operate long-term without energy replenishment. For battery-dependent maritime nodes with limited recharge opportunities, this directly contributes to prolonged network sustainability. The method also represents an attempt to integrate reinforcement learning with traditional opportunistic networks, offering a new direction for subsequent research on neighbor discovery and energy conservation.

The rest of this article is organized as follows. Section 2 briefly reviews related research. Section 3 introduces a neighbor discovery method that combines active probing and passive listening. Section 4 elaborates on the specific methodology of the proposed optimization scheme. Section 5 presents simulation validations and an analysis of the results. Finally, the paper is concluded in Section 6.

## 2. Related Works

Most of the existing neighbor discovery schemes are designed for wireless ad-hoc networks. Reference [6] introduced a method for neighbor discovery that uses Q-learning to iteratively determine the working state of nodes and the direction of directional antennas based on the approximate positions of known vessels in a fleet. Reference [7] proposes an algorithm named Panacea to achieve low latency in neighbor discovery and high energy efficiency. It optimizes network performance by considering the impacts of duty cycle and packet collisions during neighbor discovery, thereby accomplishing these two objectives. Reference [8] proposed a weighted extended Kalman filtering algorithm based on the time difference of arrival/received signal strength for neighbor discovery in an unmanned aerial vehicle (UAV) ad-hoc network with relatively stable motion states. Reference [9] accelerates the neighbor discovery process by immediately scheduling a topology information exchange after any successful pairwise neighbor detection, while employing a beam selection strategy to dynamically adjust the probability distribution across beam sectors. Reference [10] employs a C-torus arbitration system to design directional transceiver beam scanning sequences for initial neighbor discovery, followed by a reinforcement learning-based adaptive beam alignment method to reduce required time slots. This approach demonstrates superior performance in discovery rate, latency, and energy efficiency. However, there has been relatively little research on reducing the energy consumption of mobile nodes in opportunistic networks in existing neighbor discovery schemes. Based on a neighbor discovery method that combines active and passive listening, the paper controls the passive listening time and uses Q-learning for power control to reduce energy consumption during the neighbor discovery process. The study is primarily based on wireless propagation models. To enhance the practicality of the proposed neighbor discovery method, both the free-space path loss model and the two-ray ground reflection model have been incorporated into the scheme.

## 3. A Neighbor Discovery Method

Existing neighbor discovery schemes can be categorized into active probing and passive listening, depending on the operational state of nodes during the discovery process. Active probing refers to the method where nodes actively broadcast probe packets, and neighboring nodes that receive these packets respond, thereby achieving the purpose of neighbor discovery. In contrast, passive listening does not rely on broadcasting probe packets for neighbor discovery [11]. Instead, nodes listen to the activity on the channel and add the information they overhear to a neighbor list for neighbor discovery. While active probing allows for rapid and efficient neighbor discovery, sending probe packets not only consumes excessive node energy but also causes interference among nodes. Additionally, excessively occupying the channel is a waste of network resources. Although passive listening consumes less energy and does not require consideration of network conflicts, it lacks the flexibility of active probing. In situations where data packets cannot be parsed to obtain neighbor information, active probing is needed for assistance.

Reference [12] proposes a neighbor discovery scheme that combines passive listening and active probing: In opportunistic networks, nodes first passively listen to the activity on the channel. Based on whether they can parse the MAC-layer data frames of overheard nodes, nodes are classified into clear neighbor nodes and ambiguous neighbor nodes. If the ambiguous neighbor list is empty, nodes directly broadcast hello probe packets at maximum power to achieve rapid neighbor discovery. The detected nodes respond with hello_reply messages carrying their own information to inform the probing nodes of their existence. If the neighbor list is not empty, hello probe packets are broadcast using a method of incrementally increasing power until it reaches the maximum level, and the detected nodes respond accordingly [13]. This approach aims to reduce energy consumption and decrease network conflicts. The specific steps are as follows:At the initial stage of neighbor discovery, any given node listens to the activity of surrounding nodes. Based on whether it can clearly parse the data frames at the MAC layer, the node categorizes neighboring nodes into clear neighbor nodes and ambiguous neighbor nodes, and adds them to the clear neighbor list and ambiguous neighbor list, respectively.If the ambiguous neighbor list of a node is empty, the node directly broadcasts hello probe packets carrying its own power information at maximum power to initiate neighbor discovery. Neighboring nodes that receive these probe packets respond with hello_reply packets using the same power. This active probing based on maximum power allows for rapid neighbor discovery Third item.If the ambiguous neighbor list of a node is not empty, as indicated in reference [8], dividing the maximum power into five levels offers a good compromise between the frequency of new neighbor discoveries and node energy consumption. Therefore, the maximum transmission power is evenly divided into five levels. The node employs an active probing method with incrementally increasing power levels. The detected nodes continue to respond using the same power level as the probe they received. This approach helps reduce network conflicts caused by direct maximum power probing among nodes.

## 4. Method Optimization

Passive listening and active probing in neighbor discovery can be understood as a process where nodes continuously listen and transmit packets during the neighbor discovery phase. Nodes actively probe for neighbor nodes by broadcasting hello probe packets. In addition to listening for hello_reply messages from neighboring nodes, nodes also listen to other active nodes in the channel for neighbor discovery. Therefore, by setting the duration for passive listening, the advantages of passive listening can be maximized. At the same time, energy consumption can be reduced by controlling the power of active probing packets.

### 4.1. Passive Listening Time

Indicated in references [12,14], when the maximum transmission power of nodes is configured at 45 dBm and the node density ranges from 1.2 to 1.6 nodes per square kilometer, the network performance attains an optimal level. Consequently, the passive listening duration can be dynamically adjusted according to the node density. Furthermore, key performance indicators (KPIs)—including throughput, end-to-end latency, and packet loss rate—may be employed to determine the appropriate passive listening parameters for maximizing network efficiency.

### 4.2. Q-Learning Algorithm and Power Optimization Method

Q-learning reinforcement learning achieves convergence based on the Markov decision process and is also one of the commonly used algorithms of the temporal difference algorithm [15,16]. This algorithm mainly constructs an optimization strategy by mapping from the environmental state to the behavioral state [17]. Through the interaction between the agent and the environment, it continuously acquires rewards and punishments from unsupervised learning in the environment to achieve the purpose of self-learning. It mainly consists of: state policy (state), action policy (action), reward and punishment policy (reward), and optimization policy (policy). The algorithm is shown in Figure 1. In the figure, the agent generates an action policy based on the previous state and the reward and punishment policy, and continuously constructs an action policy with the environment to obtain the maximum reward value.

The specific principles of the algorithm are shown in Equations (1) and (2),(1)$$Q\left(s_{t},a_{t}\right)=Q\left(s_{t},a_{t}\right)+\beta \left[\mu +\gamma Q\left(s_{t+1},a_{+1t}\right)-Q\left(s_{t},a_{t}\right)\right]\;,$$(2)$$Q\left(s_{t},a_{t}\right)=Q\left(s_{t},a_{t}\right)+\beta \left[\mu +\gamma {max}_{a}Q(\;s_{t},a_{t}\;)Q\left(s_{t+1},a_{+1t}\right)-Q\left(s_{t},a_{t}\right)\right]\;,$$

Among them, $Q(s_{t},a_{t})$ is the action selection matrix; $s_{t}$ is the selection state; $a_{t}$ is the selection action; $s_{t+1}$ is the selected state for the next moment; $a_{t+1}$ is the selection action for the next moment; β is the learning rate of new knowledge; μ is the reward value obtained by the system; γ is the discount factor of the rewards obtained in the future; ${max}_{a}Q(s_{t},a_{t})$ is the maximum value of selecting action $a_{t+1}$ for the next state $\;s_{t+1}$. Through the derivation of the formula, it can be transformed into Equation (3),(3)$$Q\left(s_{t},a_{t}\right)=\mu +\gamma {max}_{a}Q\left(s_{t+1},a_{+1t}\right)\;,$$

From Equation (3), it can be obtained that the Q-learning algorithm is independent of the initial state. Even without obtaining the environmental model in advance, it can gradually ensure the convergence of the algorithm through iterations based on the existing experience.

The Q-learning algorithm selects the corresponding action value according to $\;{max}_{a}Q(s_{t},a_{t})$. However, if each exploration follows this strategy, it will result in a local optimal solution due to being trapped in the existing experience. Therefore, in the next action, not only the action with the maximum Q value should be selected to obtain the maximum reward value, that is, the exploitation policy; but also, other actions should be selected to innovate the best path, that is, the exploration policy. Therefore, the ε-greedy policy is adopted, as shown in Equation (4),(4)$$\mathrm{prod}\left(a_{i}\right)=\left\{\begin{array}{c} 1-\varepsilon ,a=\mathrm{argmax}Q\left(si,ai\right)\\ \varepsilon ,other \end{array}\right.\;,\;$$

In the formula, ε is the degree of greed. By setting the value of ε to balance the action selection of the “exploitation” and “exploration” strategies, the situation of being trapped in the local optimal solution can be avoided, thereby ensuring the rapid convergence of the Q matrix and achieving the optimum.

Q-learning does not require prior knowledge of the complete model of the environment. It can learn the optimal strategy through interaction and trial-and-error with the environment [18]. The proposed approach exhibits robustness in dynamic and uncertain environments by adaptively updating its strategy in response to evolving reward signals and state transitions. Consequently, Q-learning is well-suited for dynamically optimizing the transmission power for active detection, as it efficiently adjusts to environmental variations.

In the combined passive and active neighbor discovery scheme, the stepwise increasing active probing method simply divides the maximum transmission power into five levels, making the energy consumed in each active probing certain, and the neighbor nodes also objectively exist in the network scenario. Aiming at detecting as many neighbor nodes as possible with limited energy, when performing active probing with increasing power, the interval of each power level is evenly divided into three equal parts, giving each level a floating space, and using Q-learning reinforcement learning to select the power with the lowest average energy consumption of the neighbor nodes detected by the node in each level for neighbor discovery [16]. Taking Figure 2 as an example, the black arc indicates that the maximum transmission power of the node is divided into two levels, the green dotted arc indicates the up and down floating selection of the first-level power, and $P_{11},P_{12},P_{13}$ denote the available power levels under the primary power grade. The red dotted arc indicates the up and down floating selection of the second-level power, $P_{21},P_{22},P_{23}$ represent the selectable power levels under the secondary power classification.

### 4.3. Neighbor Expiration Detection

If a node does not receive any messages from a neighboring node within its neighbor expiration time, it may be inferred that the neighboring node is no longer within its transmission range, and the neighboring node can thus be removed from the neighbor list. In this scheme, each node is equipped with two timers: a hello timer and a neighbor timer. The hello timer triggers the transmission of a hello message upon expiration, while the neighbor timer is responsible for periodically purging expired neighbor entries. The neighbor expiration timeout is defined as Equation (5),(5)$$t_{expried}=\left(Allowed\_Hello\_Loss+1\right)*t_{hello}+t_{delay},\;$$

In Equation (5), Allowed_Hello_Loss denotes the maximum number of consecutive hello message losses permitted. This value typically ranges between 0 and 2. A value of 0 indicates an ideal network environment with no packet loss, though such conditions are relatively uncommon in practice. The parameter t_hello_ represents the transmission interval of hello messages. The neighbor expiration timeout must be set to a value greater than this hello transmission interval. The parameter t_delay_ is introduced to delay the expiration of neighbor entries. Its primary function is to prevent premature neighbor removal and to account for potential delays in hello message reception caused by signal interference or transient connectivity issues. In this scheme, t_delay_ is tentatively set to 0.5 s. Given that wireless opportunistic networks generally operate in challenging environments, the value of Allowed_Hello_Loss is set to 2. Since the hello transmission interval is not a core focus of this study, it is fixed at 1 s in this implementation. Consequently, the neighbor expiration time t_expired_ is 3.5 s.

### 4.4. Radio Propagation Model

The free-space model represents the most fundamental wireless propagation model, which assumes an infinite vacuum between transmitter and receiver without any obstacles. In this model, the transmission range of a node forms a perfect circular region centered at the node itself. Only nodes located within this circular area can receive signals from the transmitting node. The received power is given by Equation (6),(6)$$P_{r}=\frac{P_{t}G_{t}G_{r}\lambda^{2}}{{\left(4\pi \right)}^{2}d^{2}L},$$
where,$G_{t}$ and $G_{r}$ represent the antenna gains of the transmitting node and receiving node, respectively. d denotes the propagation distance between nodes, L accounts for system loss factors independent of transmission, λ indicates the carrier wavelength.

In the free-space propagation model, the path loss (PLPL) can be mathematically expressed as Equation (7),(7)$$PL\left(\mathrm{dB}\right)=10\log\frac{P_{t}}{P_{r}}=-10\log\frac{G_{t}G_{r}\lambda^{2}}{{\left(4\pi \right)}^{2}d^{2}},$$

If antenna gains are not considered, substituting the λ=c/f into the Equation (6), then, derives the following Equation (8),(8)$$PL\left(\mathrm{dB}\right)=32.4+20\mathrm{lg}f\left(\mathrm{MHz}\right)+20\mathrm{lg}d\left(\mathrm{km}\right),$$

The two-ray ground reflection model accounts for both the direct line-of-sight propagation path and the ground-reflected path, making it more suitable for longer-range propagation scenarios. The received signal power in this model is given by Equation (9),(9)$$P_{r}=\frac{P_{t}G_{t}G_{r}{h_{r}}^{2}{h_{t}}^{2}}{d^{4}L},$$

In the formula, $h_{t}$ and $h_{r}$ denote the heights of the transmitting and receiving antennas, respectively. Compared with Equation (6), this propagation model exhibits faster power attenuation with distance, The propagation distance can be expressed as Equation (10),(10)$$d_{cross}=\frac{{4\pi h}_{t}h_{r}}{\lambda },$$

In the proposed scheme, the appropriate propagation model is selected based on the distance D between the transmitting and receiving nodes. When D < $d_{cross}$, using the free-space model, when D ≥ $d_{cross}$, using the two-ray ground reflection model.

Therefore, the improved algorithm flow is as follows:

The node passively listens to the channel for a period of time, and divides the neighbor nodes into clear neighbor nodes and fuzzy neighbor nodes by whether the MAC layer data frame can be clearly parsed, and adds them to the clear neighbor node set and the fuzzy neighbor node set, respectively.

If the fuzzy neighbor list is empty, broadcast the probe packet at the maximum power Pm directly for neighbor discovery.If the fuzzy neighbor list is not empty, take P_m_/5 − P_m_/15, P_m_/5, and P_m_/5 + P_m_/15 as the initial transmission power, respectively, and take P_m_ − P_m_/15, P_m_, and P_m_ + P_m_/15 as the maximum transmission power, respectively, and perform three times of active probing with increasing power step by step with an increasing power value of P_m_/5. Record the number of neighbor nodes ${\;n}_{ij}$ detected each time, and calculate the average power consumption of the detected nodes ${\bar{P}}_{ij}\;$=${\;P}_{ij}$/$n_{ij}$, complete the initialization. Where i is the power level selection (i = 1, 2, 3, 4, 5), and j is the up and down floating selection of the power at this level (j = 1, 2, 3). Therefore, Equation (10) can be obtained,(11)$$P_{ij}=i(P_{m}/5)+(j-2)(P_{m}/15)\;,\;$$ In order to save the energy consumed in neighbor discovery, the smaller the average power consumption of the detecting node, the better. During the subsequent iterative process, there is a preference for selecting a power level with a smaller ${\bar{P}}_{ij}$. Therefore, $1/{\bar{P}}_{ij}$ can be used as the reward value. The power level with a smaller average power consumption will obtain a greater reward value. The node constructs the initialization matrix $Q\left(s,\;a\right)$ and the initialization iteration period.

3.When the node performs active probing with increasing power again, when the power level is $P_{\cdot j}$, the power with the maximum ${\bar{P}}_{\cdot j}$ is selected for active probing with a probability of 1–ε, and other powers are selected with a probability of ε for active probing, and the current ${\bar{P}}_{ij}$ value is updated simultaneously.

The flow chart is shown in Figure 3.

However, unlike the traditional Q-learning reinforcement learning, in this algorithm, since the reward value also changes accordingly with the movement of the nodes in the scenario, the Q matrix will not converge, but precisely because of this, the appropriate power can be dynamically selected for neighbor discovery.

## 5. Simulation and Result Analysis

### 5.1. Simulation Setup

The simulation experiments in this study were conducted using the Exata 5.1 platform, a high-fidelity network emulator that enables rapid and accurate performance evaluation. Exata employs a software-defined virtual network to digitally model the entire network architecture, including protocol stacks, antenna configurations, and network devices. In this experiment, the passive listening duration was initially optimized based on node density [19], and then observes the changes in indicators such as throughput and energy consumption in the network scenario in the power control neighbor discovery algorithm based on Q-learning reinforcement learning to test the performance of the algorithm.

In order to reflect the sparse characteristics of wireless opportunistic network nodes, a 5000 m × 5000 m network scenario is selected and 15 mobile nodes are placed in the scenario. The node moving speed is between 0 and 30 m/s. The specific planar simulation scenario is shown in Figure 4. The basic scenario parameter settings are shown in Table 1.

The following is the analysis of the simulation experiment results.

### 5.2. Simulation Results

By setting different passive listening times, compare the differences in network indicators in the scenario.

In this experiment, the maximum transmission power of the node is 45 dBm, and other parameter settings are the same as in Table 1. Four sets of control experiments are set up. It can be seen from Figure 5a that as the passive listening time continues to increase, the throughput of the entire scenario shows a downward trend [20]. Mainly because for the 5000 m × 5000 m network scenario, arranging 15 nodes makes the nodes in the entire network very sparse. At this time, even if the node has a large transmission power, it will not cause too many data packets in the network to cause conflicts. With the increase of passive listening time, it will instead cause the frequency of node activities in the channel to decrease, the speed of neighbor discovery to slow down, and the throughput of the entire scenario to decrease. Therefore, for a sparse network scenario, setting a shorter passive listening time is helpful for quick neighbor discovery.

In the 5000 m × 5000 m scenario, in order to test the appropriate time of passive listening under the optimal node density of the combined passive and active neighbor discovery algorithm, the number of nodes in the scenario is increased to 40, other parameters remain unchanged, and still four sets of control experiments are set. As shown in Figure 5b, the throughput of the entire scenario reaches the highest at 0.4 s. This phenomenon primarily occurs because, in the 5000 m × 5000 m network scenario, the node density leads to frequent packet transmissions, driving the network into a near-saturation state. A shorter passive listening duration results in excessive concurrent packet transmissions, which escalates channel contention and network congestion. Consequently, the average network throughput experiences a measurable degradation under these conditions. While setting a longer passive listening time, the situation of conflicts caused by too many data packets in the network will decrease, but at the same time, the frequency of node activities in the channel decreases, and the throughput will inevitably decrease.

Through the above analysis, calculated according to the optimal maximum transmission power of 45 dBm of the algorithm, the node coverage is a circular range of about 7.6 km^2^ centered on itself, and at this time, the neighbor nodes of the node are about 10, making the network quality of the neighbor area where the node is located reach a peak. Therefore, it can be stipulated that the passive listening time of the node is 0.1 s. If the node detects 10 or more neighbor nodes, the passive listening time of the node is correspondingly extended to 0.4 s. However, it should be noted that the passive listening times of 0.1 s and 0.4 s are the more appropriate values selected through the comparison experiment. The following experiments are temporarily set as this passive listening time for now.

Comparison of network indicators of the combined passive and active neighbor discovery algorithm and the Q-learning power control algorithm. To ensure algorithmic stability by preventing excessive parameter fluctuations while accounting for the significance of long-term returns—thereby mitigating short-term decision bias induced by an overemphasis on distant rewards—the experimental parameters were set as follows: β = 0.1, γ = 0.8, ε = 0.2.

This experiment sets different maximum transmission powers, and other parameter settings are the same as in Table 1. Figure 6 illustrates the impact of transmission power on network throughput. As transmission power increases, the average system throughput exhibits a corresponding rise. When the maximum transmission power remains below 35 dBm, the throughput difference between the two algorithms is negligible. However, once the power reaches 35 dBm, the Q-learning-based power control algorithm demonstrates a modest throughput improvement, achieving an average gain of approximately 3%. This phenomenon can be explained by the expanded discovery range at higher transmission powers, which facilitates the detection of a larger number of neighboring nodes. The resulting improved delay management reduces inter-node packet collisions, thereby contributing to the observed throughput enhancement. Nevertheless, compared to the conventional algorithm, the reinforcement learning (RL)-based approach does not yield a statistically significant throughput improvement under these conditions.

As shown in Figure 7a,b, as the transmission power of the node increases, its search range expands correspondingly, leading to a continuous rise in both the average number of neighbor discoveries and the associated energy consumption. Both algorithms employ a cross-layer cooperative neighbor discovery approach, integrating active physical-layer power control probing with passive media access layer (MAC) listening. Transmission power is dynamically optimized through a cross-layer optimization framework, enhancing efficiency through reinforcement learning. This method achieves superior neighbor discovery performance, detecting a larger number of neighboring nodes while minimizing energy expenditure.

The proposed RL-based algorithm demonstrates a significant reduction in average energy consumption per neighbor discovery compared to conventional methods. Moreover, it consistently outperforms legacy schemes in terms of average neighbor discovery rate. When the transmission power remains below 35 dBm, the energy consumption difference between the two schemes is marginal. However, as the maximum transmission power increases further, the RL-based algorithm exhibits substantially improved energy efficiency. At 50 dBm, the proposed method achieves approximately 29% energy savings compared to traditional approaches.

As shown in Figure 8, the average energy consumption of neighbor discovery also increases with the increase of transmission power. Since the average number of neighbor discoveries and the transmission power are basically in a linear relationship, and the average energy consumption and the transmission power are basically in an exponential relationship, the average energy consumption of neighbor discovery also basically shows an exponential relationship. It can be seen from Figure 8 that when the transmission power is 40, 45, and 50 dBm for neighbor discovery, the energy savings are more obvious, and the average energy consumption is reduced by about 22%, 33%, and 25% respectively. This also indicates that when the maximum transmission power is 45 dBm for neighbor discovery, the cost performance of energy consumption is the highest. It can also be seen that when the transmission power is at a relatively low level, the effects of the two methods are not significantly different; when the transmission power is at a relatively high level, the reinforcement learning algorithm shows better performance. This is mainly because the nodes in the wireless opportunistic network scenario are sparse. A relatively low transmission power results in a relatively small number of detected nodes, and the method of power level division has little impact on relatively low transmission powers. This makes the results of the reinforcement learning method, which selects a power with a relatively low average energy consumption for neighbor discovery, not very different from those of the old method. When a relatively high transmission power is selected, the nodes can detect more neighbor nodes, and the power division of the reinforcement learning method is more distinct. Since the reinforcement learning method selects a power with a lower average energy consumption for neighbor discovery to operate, this method shows better performance at relatively high transmission powers.

From Figure 9, it can be observed that the end-to-end delay exhibits irregular fluctuations as the transmission power of nodes in the network increases. When the transmission power is set to 20 dBm, the end-to-end delay remains relatively low. This is primarily because the active probing power is minimal, resulting in limited opportunities for data transmission among nodes in the network scenario. When transmission does occur, it is mostly completed within a single hop, leading to a lower delay for successful transmissions. In the range of 25–40 dBm, the transmission range of nodes expands with increasing power, making data transmission more complex. In this phase, data may be transmitted either within a single hop or through multiple hops. Additionally, due to the unpredictable mobility of nodes in opportunistic networks, the variation in end-to-end delay becomes more intricate—neither simply increasing nor decreasing—though the overall trend shows a reduction. When the power exceeds 40 dBm, the active probing power becomes sufficiently high, allowing nodes to largely avoid communication interruptions caused by mobility. Consequently, data transmission is predominantly completed within a single hop, resulting in minimal end-to-end delay. Figure 9 further reveals that the two neighbor discovery methods exhibit nearly identical trends in end-to-end delay variation. Moreover, neither method demonstrates a significant advantage in optimizing end-to-end delay performance.

As shown in Figure 10, the packet loss rate in the network scenario generally decreases as the transmission power increases. This is primarily because a higher transmission power expands the communication range of nodes, thereby improving network connectivity. Consequently, fewer packets are lost due to routing failures, leading to a gradual reduction in the packet loss rate. However, when the maximum transmission power is below 35 dBm, the conventional method exhibits a lower packet loss rate compared to the Q-learning-based scheme. This discrepancy arises because, under low transmission power conditions, the reinforcement learning approach introduces greater uncertainty in power adjustment, and node mobility further exacerbates connection instability. In contrast, when the maximum transmission power exceeds 35 dBm, the difference in packet loss rates between the two methods becomes negligible. This suggests that the proposed Q-learning scheme is more suitable for scenarios with higher transmission power.

### 5.3. Discussion

The experimental results show that by controlling the time of passive listening and using Q-learning reinforcement learning to control the power of active detection, the overall throughput in the network scene of opportunistic network neighbor discovery is improved, the number of neighbor discovery also has a certain increase, which reduces the energy consumption for neighbor discovery and improves the average discovery rate. However, Q-learning reinforcement learning algorithm in the end-to-end delay and packet loss rate does not show any improvement.

This neighbor discovery method still faces challenges such as high energy consumption during the initialization phase, as well as excessive network overhead and energy waste caused by either too many or too few neighboring nodes. In the initial stage, due to the lack of prior knowledge about surrounding nodes, the method requires 15 rounds of active probing to complete initialization. Although energy consumption is reduced in subsequent phases, the unavoidable overhead during initialization contradicts the core goal of energy efficiency, highlighting an area that requires further improvement. While the method is suitable for sparse maritime opportunistic networks, its real-world application may encounter issues such as network conflicts due to an excessive number of neighbors, or energy waste when no neighbors are present. These practical limitations indicate the need for continued refinement to enhance robustness and adaptability.

## 6. Conclusions

In network environments with dynamically changing node densities, adaptive passive listening duration configuration can effectively mitigate channel contention issues, particularly demonstrating significant advantages in high-density scenarios. By dynamically adjusting the listening duration based on real-time node density, the packet collision rate is substantially reduced. This study further employs a Q-learning-based reinforcement learning approach to optimize the neighbor discovery process by selecting transmission power levels that minimize the average energy consumption of active probing. During simulations, appropriate propagation models were applied according to the distance between source and receiver nodes, making the results more aligned with real-world conditions and enhancing the validity of the conclusions. Simulation results indicate that the proposed reinforcement learning-based power control algorithm improves throughput by up to 3%, increases the average number of discovered neighbors, and reduces energy consumption by up to 29%. At a maximum probing power of 45 dBm, the average energy consumption for neighbor discovery was decreased by approximately 33%, demonstrating a significant improvement in energy-efficient neighbor discovery performance in opportunistic networks.

Furthermore, the passive listening duration configuration scheme proposed in this study requires further controlled experiments to determine optimal parameter settings. Future work could integrate the listening duration as a learnable parameter within the reinforcement learning framework to enable dynamic adaptation and enhance overall network performance. Given the highly dynamic nature of the maritime environment, subsequent research should also incorporate factors such as harsh weather conditions and utilize marine communication channel models under various natural scenarios. This would strengthen the applicability and robustness of the method for neighbor discovery in maritime internet environments. While the current implementation uses the AODV routing protocol—which shares certain characteristics with opportunistic networking—future studies could introduce standard opportunistic protocols (such as PRoPHET or Spray and Wait) in the Exata platform to support more robust and systematic performance evaluation and improvement. Since the proposed method emphasizes discovering as many neighbor nodes as possible with low power consumption, it may detect nodes that contribute minimally to immediate data transmission. However, such neighbor information could still be utilized to enrich routing decisions and enhance path diversity in delay-tolerant maritime networks.

## Figures and Tables

**Figure 1 sensors-25-05720-f001:**
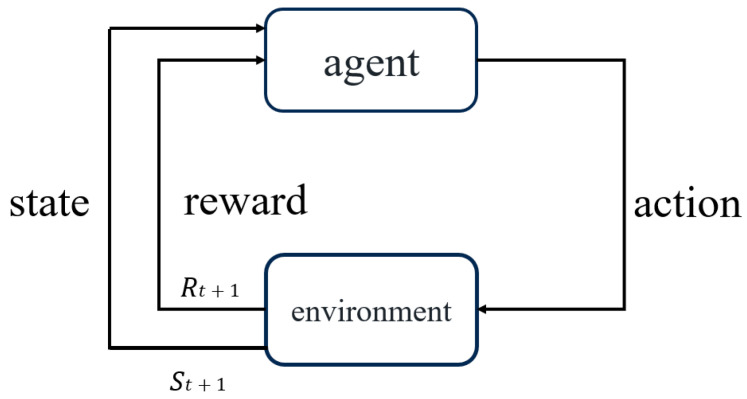
Block diagram of Q-learning algorithm.

**Figure 2 sensors-25-05720-f002:**
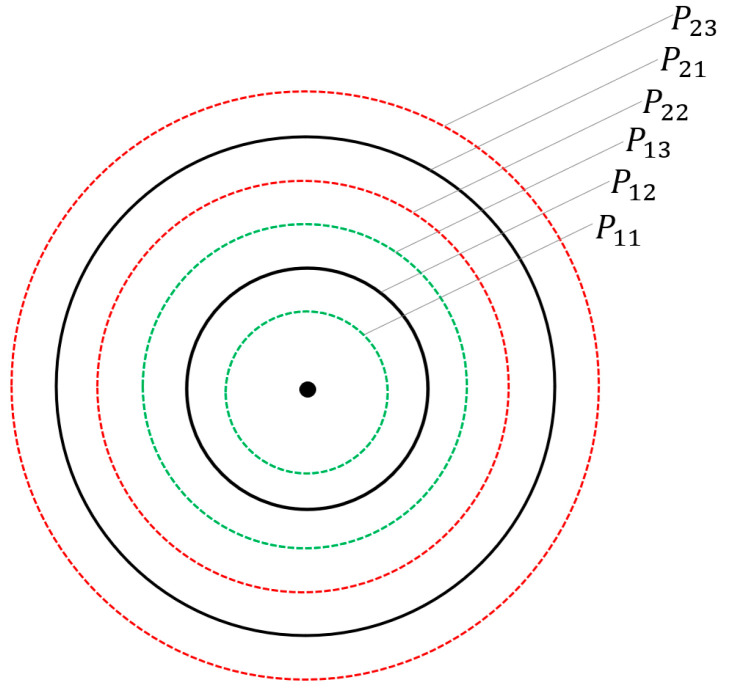
Schematic diagram of level division.

**Figure 3 sensors-25-05720-f003:**
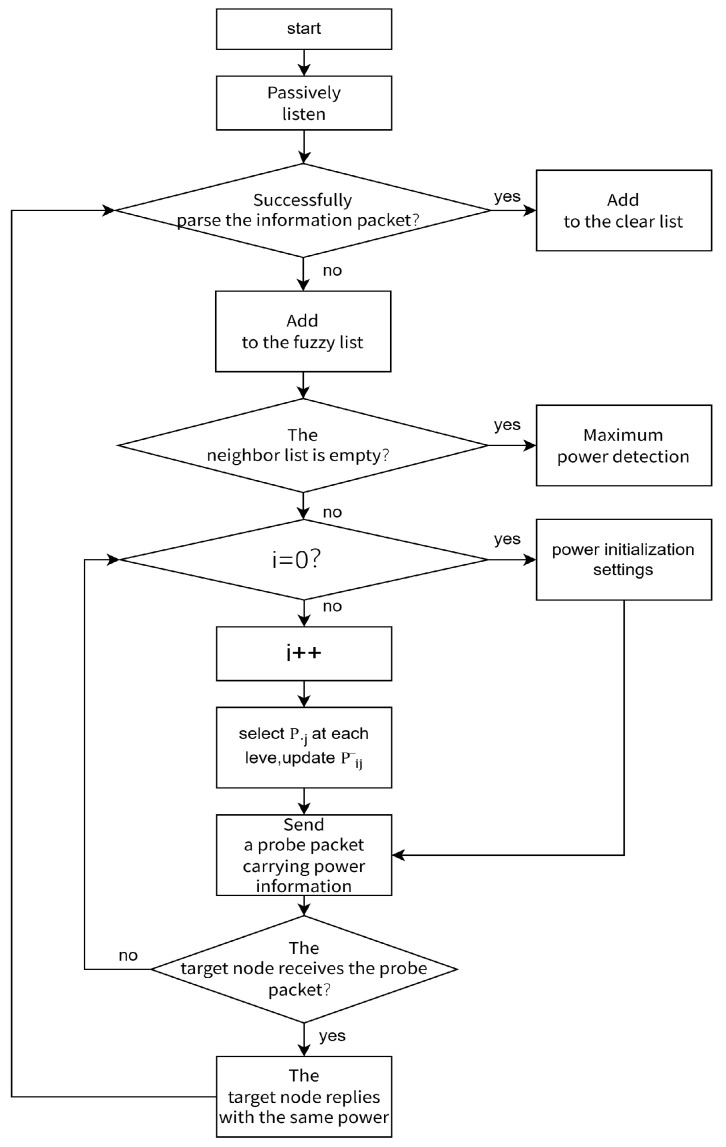
Algorithm flow chart.

**Figure 4 sensors-25-05720-f004:**
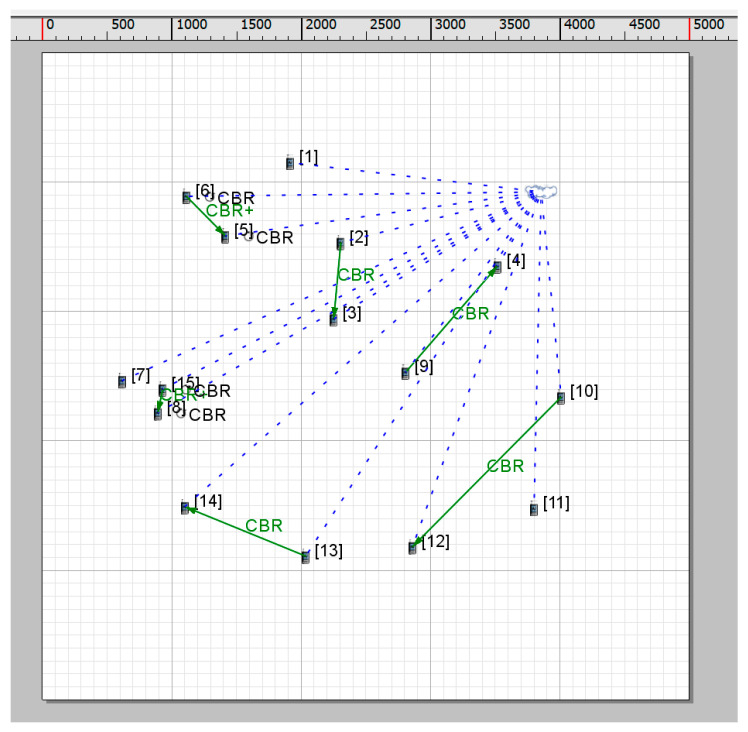
Simulation scenario of power control neighbor discovery algorithm.

**Figure 5 sensors-25-05720-f005:**
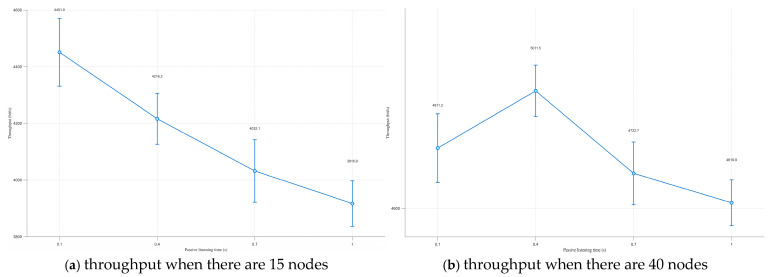
The influence of listening time on throughput when there are different nodes.

**Figure 6 sensors-25-05720-f006:**
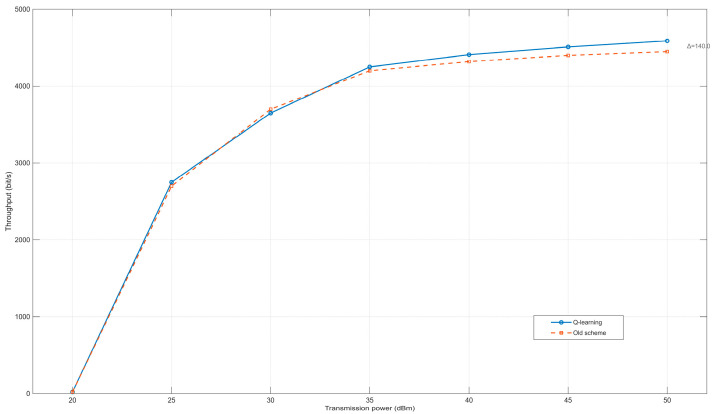
Relationship between transmission power and throughput.

**Figure 7 sensors-25-05720-f007:**
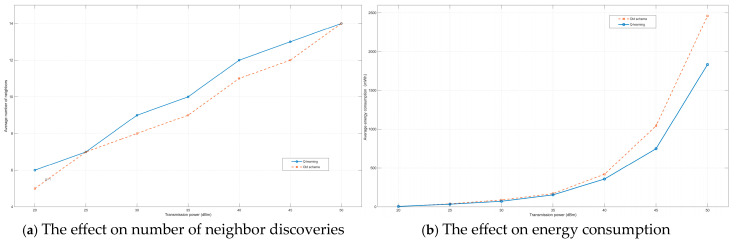
Comparison of the effects of two methods on number of neighbor discoveries and energy consumption under different transmission powers.

**Figure 8 sensors-25-05720-f008:**
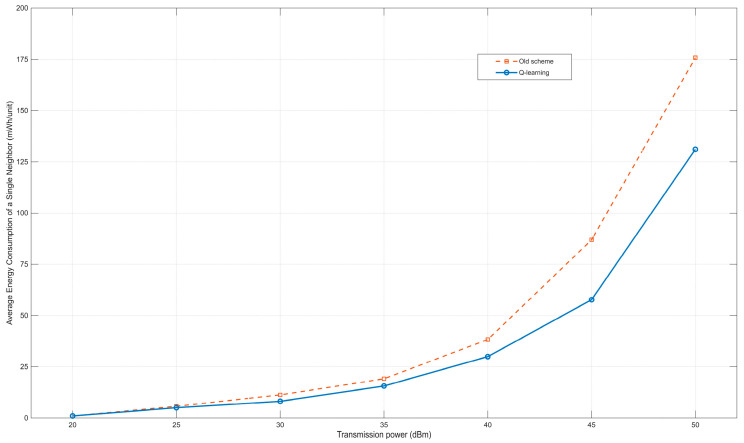
Relationship between transmission power and average power consumption for single neighbor discovery.

**Figure 9 sensors-25-05720-f009:**
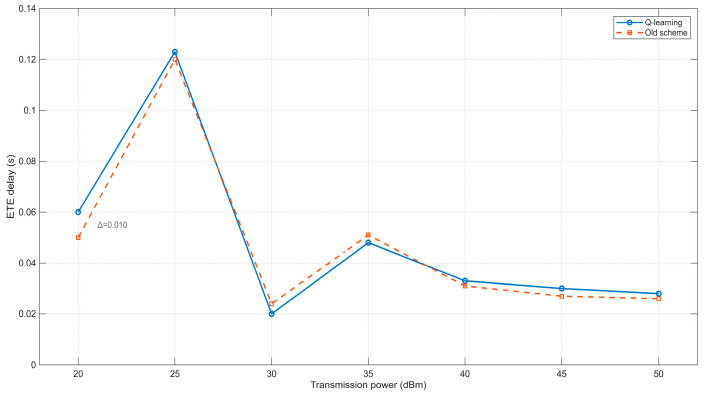
Relationship between transmission power and ETE delay.

**Figure 10 sensors-25-05720-f010:**
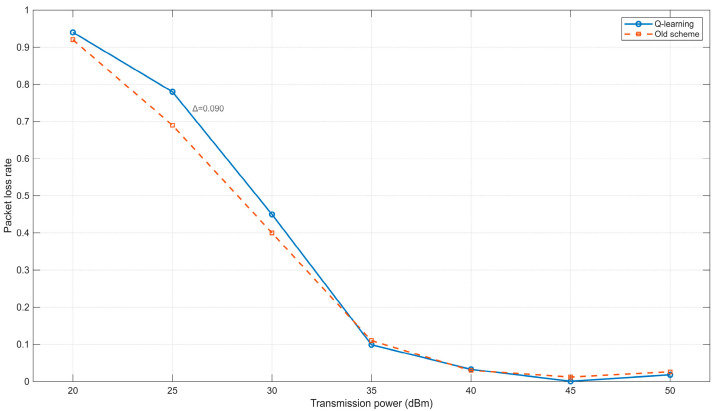
Relationship between transmission power and Packet loss rate.

**Table 1 sensors-25-05720-t001:** Scenario parameter settings.

Parameters	Values
Scenario area/m^2^	5000 × 5000
Simulation time/s	3600
Number of nodes	15
Node movement model	Random Waypoint
Power consumption for sending packets/mW	100
Power consumption for receiving packets/mW	130
Idle power consumption/mW	120
Node energy model Generic	Generic
MAC layer protocol	802.11
Network layer protocol	IPv4
Transport layer protocol	UDP
Application data	CBR
Packet size/bytes	512
Temperature/K	290

## Data Availability

The program code used in the research can be obtained from the corresponding author upon reasonable request.

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
