# Peer review of "A Q-Learning Based Scheme for Neighbor Discovery and Power Control in Marine Opportunistic Networksâ€"

_sensors, 2025, doi:10.3390/s25185720_

Round 1
Reviewer 1 Report
Comments and Suggestions for Authors
I have carefully reviewed efforts carried out in their proposed opportunistic networks. While the work seems interesting, there are the major practical limitations of the proposed neighbour discovery method described in the work, based on both its theoretical assumptions and real-world constraints:
- In Section 4.2, and other places, the authors failed to account for high computational and memory overhead. As you can see, the use of Q-learning for transmission power optimisation introduces non-trivial computational and memory costs. The consequence is that low-power devices typical in opportunistic networks (e.g., smartphones, wearables, sensors), maintaining and updating Q-tables in real time may exceed hardware capabilities, especially in dynamic topologies. This was not addressed!
- There is absolute lack of convergence in q-learning in Figure 3. In fact, as explicitly stated in the paper, the Q-matrix does not converge due to the constantly changing neighbour set (because of node mobility). This undermines one of Q-learning’s key strengths (convergence to optimal policy), leading to instability and potentially suboptimal or oscillatory power level decisions.
- The authors overlooked the overhead of multi-level power probing in Section 4.2. Essentially, dividing the maximum power into five levels with three sub-levels each result in 15 probing rounds in the worst-case scenario. In my view, this significantly increases the number of transmissions, consuming more energy and possibly negating the intended efficiency gains. This is another major issue!
- Also, There is a serious case of static Power Segmentation which ignores Hardware Variability. In the work, the segmentation of power levels is uniformly applied (e.g., Pm/5, Pm ± Pm/15) regardless of actual radio transceiver capabilities. However, in real hardware, power levels are often discrete and non-linear, making such granular control inapplicable or ineffective in practice. This was not addressed.
- I noted the imprecise passive listening heuristics in the work. Now, passive listening relies on the node’s ability to parse MAC frames to distinguish between “clear” and “ambiguous” neighbours. This assumes stable and interference-free channels, but in noisy or encrypted networks, MAC frames may be inaccessible, leading to misclassification or missed discoveries. This is a major issue!
- There is this dependence on accurate node density estimation. The adaptive passive listening time is based on node density assumptions (e.g.,1.2–1.6 nodes/km²). Opportunistic networks often have extreme variability in node density, making it hard to set an optimal value without prior or real-time knowledge, leading to either energy waste or incomplete discovery. This is a major issue!
- In Section 5, there is this concern around Mobility Model Mismatch in that the simulations use idealised mobility (0–30 m/s uniform random) without accounting for real-world behaviours like pause time, group mobility, or social patterns. This would mean that the performance gains shown in simulation may not generalise to human-centric or vehicular opportunistic networks.
- There is a clear case of hidden terminal and collisions During Probing. In fact, even with incrementally increasing probing power, the lack of collision avoidance mechanisms (e.g., CSMA/CA) means that hello and hello-reply to packets may collide, especially in overlapping discovery zones. Essentially, this reduces discovery accuracy and increases retransmissions, raising energy consumption.
- The authors did not consider Message Buffering or Delay Tolerance. Opportunistic networks often operate in a delay-tolerant fashion, where discovering a neighbour does not immediately imply successful message exchange. The algorithm optimises neighbour discovery but not contact utility, potentially discovering many neighbours who may not be useful for actual data forwarding.
- Simulation scope is simply narrow in Section 5. The simulation only tests a single 5000m x 5000m scenario with 15 nodes, limiting the generalisation of results really. This ignores practical conditions such as urban obstacles, varying node densities, or larger-scale deployments.
- While the hybrid passive-active + Q-learning scheme is innovative in theory, its real-world viability is limited by energy, computation, and protocol stack constraints inherent to opportunistic and delay-tolerant networks. This work will benefit from reduction in algorithmic complexity; Incorporate social/temporal utility into discovery; Integrate with realistic MAC protocols; Extend simulations to diverse environments.
- The research gaps is vague, and no clear contribution summary was set out. Authors should confirm how their proposal is different from these works
- A novel Q-learning-based secure routing scheme with a robust defensive system against wormhole attacks in flying ad hoc networks - ScienceDirect
- A Joint Optimization Design Method for Opportunistic Network Neighbour Discovery and Power Control Based on Q-Learning | IEEE Conference Publication | IEEE Xplore
Comments on the Quality of English Language
Needs improvements
Author Response
Comment 1:In Section 4.2, and other places, the authors failed to account for high computational and memory overhead. As you can see, the use of Q-learning for transmission power optimisation introduces non-trivial computational and memory costs. The consequence is that low-power devices typical in opportunistic networks (e.g., smartphones, wearables, sensors), maintaining and updating Q-tables in real time may exceed hardware capabilities, especially in dynamic topologies. This was not addressed!
Response 1:Thank you for pointing this out. I agree with this comment.My research focuses on maritime internet networks. Since most networks in the marine environment operate as opportunistic networks, I broadly refer to them as such in my writing. However, I will confine the scope of this study specifically to maritime internet applications. In this context, network nodes primarily consist of large-scale entities such as ships.Additionally, regarding the memory overhead, I have now incorporated the neighbor table update method into the article. Previously, I did not include it as I considered it less critical, but its addition further as 4.3 enhances the completeness of the paper.
Comment 2:There is absolute lack of convergence in q-learning in Figure 3. In fact, as explicitly stated in the paper, the Q-matrix does not converge due to the constantly changing neighbour set (because of node mobility). This undermines one of Q-learning’s key strengths (convergence to optimal policy), leading to instability and potentially suboptimal or oscillatory power level decisions.
Response 2:Thank you for pointing this out. I fully acknowledge the reviewer’s insightful concerns. Indeed, in dynamic maritime opportunistic networks, the constant changes in neighbor sets due to node mobility prevent traditional Q-learning from guaranteeing strict mathematical convergence, which may lead to power decision fluctuations or suboptimal solutions.However, our objective is not to pursue theoretical convergence in the traditional Q-learning sense but rather to leverage its online adaptation capability for dynamic environments. Although the Q-matrix does not achieve complete convergence, the learning algorithm can still maintain network performance within an acceptable range through continuous policy adjustments.
Comment 3:The authors overlooked the overhead of multi-level power probing in Section 4.2. Essentially, dividing the maximum power into five levels with three sub-levels each result in 15 probing rounds in the worst-case scenario. In my view, this significantly increases the number of transmissions, consuming more energy and possibly negating the intended efficiency gains. This is another major issue!
Response 3:Thank you for pointing this out. I acknowledge that during the initial neighbor discovery phase, fine-grained power probing may indeed incur significant energy overhead and network costs due to the high number of attempts. However, in subsequent phases, the Q-learning-based power control method demonstrates substantial long-term energy savings. While the initial neighbor discovery stage consumes more energy, subsequent phases exhibit markedly reduced energy expenditure.I sincerely appreciate your valuable feedback. This is an important observation, and I will further investigate this aspect in future research to optimize the trade-offs between initial overhead and long-term efficiency.
Comment 4:Also, There is a serious case of static Power Segmentation which ignores Hardware Variability. In the work, the segmentation of power levels is uniformly applied (e.g., Pm/5, Pm ± Pm/15) regardless of actual radio transceiver capabilities. However, in real hardware, power levels are often discrete and non-linear, making such granular control inapplicable or ineffective in practice. This was not addressed.
Response 4:Thank you for pointing this out.I fully acknowledge the point that the uniform segmentation-based power control model overlooks the discrete and nonlinear characteristics of actual RF hardware power levels. The primary reason for adopting this idealized model is to clearly validate the core mechanism of Q-learning for adaptive power decision-making within the complex dynamics of maritime networks, serving as a proof of concept.Moreover, even when applied in real-world scenarios where hardware power levels exhibit nonlinearity, the fundamental logic of the Q-learning algorithm remains unchanged. The agent continues to explore and learn to identify contextually appropriate power levels, achieving the goal of discovering more neighbor nodes with lower energy consumption.
Comment 5:I noted the imprecise passive listening heuristics in the work. Now, passive listening relies on the node’s ability to parse MAC frames to distinguish between “clear” and “ambiguous” neighbours. This assumes stable and interference-free channels, but in noisy or encrypted networks, MAC frames may be inaccessible, leading to misclassification or missed discoveries. This is a major issue!
Response 5:Thank you for pointing this out.The core challenge of the passive listening mechanism lies in whether a node can successfully decode the MAC frame information from its neighbors to achieve neighbor discovery. Its effectiveness indeed relies on stable channel conditions. Therefore, the wireless propagation model described in Section 4.4 was selected to simulate the maritime environment as realistically as possible. Simulation results demonstrate that this neighbor discovery method is well-suited for sparsely distributed node environments. Given that nodes in maritime opportunistic networks are typically deployed with low density, this approach is particularly suitable for neighbor discovery under such conditions.
Comment 6:There is this dependence on accurate node density estimation. The adaptive passive listening time is based on node density assumptions (e.g.,1.2–1.6 nodes/km²). Opportunistic networks often have extreme variability in node density, making it hard to set an optimal value without prior or real-time knowledge, leading to either energy waste or incomplete discovery. This is a major issue!
Response 6:I sincerely thank the reviewer for this insightful observation. You rightly pointed out that the listening mechanism relies on an estimated node density, which is indeed difficult to accurately obtain in dynamic maritime opportunistic networks.During the initial neighbor discovery phase, a longer passive listening period is adopted precisely because nodes lack prior knowledge of their surroundings—this helps avoid missing neighbor announcements. However, in subsequent operations, nodes adapt their passive listening duration based on learned environmental conditions.If a node is in a state with no detectable neighbors, I still opt for a shorter—yet non-zero—listening interval to prevent potentially missing newly arriving nodes. In such cases, some energy consumption is unavoidable. There is an inherent trade-off between energy efficiency and neighbor discovery completeness, and a balance must be struck between the two.Thank you once again for your valuable feedback.
Comment 7:In Section 5, there is this concern around Mobility Model Mismatch in that the simulations use idealised mobility (0–30 m/s uniform random) without accounting for real-world behaviours like pause time, group mobility, or social patterns. This would mean that the performance gains shown in simulation may not generalise to human-centric or vehicular opportunistic networks.
Response 7:Thank you for pointing this out.I sincerely apologize for my oversight. I should have clearly confined the experimental scope to my research focus—maritime internet networks. Given that nodes in the marine environment are predominantly mobile, the proposed neighbor discovery scheme is specifically designed for and more suitable to maritime internet scenarios. It is not intended for generalization to human-centric or vehicular opportunistic networks.
Comment 8:There is a clear case of hidden terminal and collisions During Probing. In fact, even with incrementally increasing probing power, the lack of collision avoidance mechanisms (e.g., CSMA/CA) means that hello and hello-reply to packets may collide, especially in overlapping discovery zones. Essentially, this reduces discovery accuracy and increases retransmissions, raising energy consumption.
Response 8:Thank you for pointing this out.The communication system selected in this paper is based on the 802.11 series network interface cards. Since the core medium access control mechanism of the 802.11 communication system is CSMA/CA, it inherently provides a certain level of collision avoidance functionality. Moreover, due to the sparse nature of nodes in opportunistic networks, the probability of packet collisions is relatively low. However, as also noted in the conclusion section of the paper, an excessive number of nodes in the network could lead to significant communication overhead, which remains an aspect requiring further improvement. Thank you once again for your valuable suggestion.
Comment 9:The authors did not consider Message Buffering or Delay Tolerance. Opportunistic networks often operate in a delay-tolerant fashion, where discovering a neighbour does not immediately imply successful message exchange. The algorithm optimises neighbour discovery but not contact utility, potentially discovering many neighbours who may not be useful for actual data forwarding.
Response 9:Thank you for pointing this out.Opportunistic networks indeed operate in a delay-tolerant manner. When a source node intends to send data to a destination node, it does not require a direct or continuous connection between them. Instead, the data can be delivered through multiple intermediate nodes via store-carry-and-forward until it eventually reaches the destination. Due to the mobility of nodes, contact opportunities are random, making the delivery time uncertain.In this context, our approach prioritizes discovering as many neighbor nodes as possible with minimal energy consumption. Even though some detected neighbors may not contribute to immediate data transmission, they expand future routing options and enhance network-layer robustness. I greatly appreciate your valuable suggestions, which provide meaningful directions for follow-up experiments and future research. These points will also be incorporated into the final discussion and conclusion of the paper.
Comment 10:Simulation scope is simply narrow in Section 5. The simulation only tests a single 5000m x 5000m scenario with 15 nodes, limiting the generalisation of results really. This ignores practical conditions such as urban obstacles, varying node densities, or larger-scale deployments.
Response 10:Thank you for pointing this out. In the simulation experiments, the absence of obstacles was intentionally designed to replicate the open network environment typical of maritime settings. The passive listening duration strategy was evaluated in a 5000m × 5000m network scenario with 15 and 40 nodes, respectively. The results demonstrate that the proposed neighbor discovery method is particularly suitable for sparse opportunistic networks. Given that node density in maritime environments is generally low, this approach is well-adapted to neighbor discovery in marine internet applications.The 15-node scenario was specifically configured to simulate extremely sparse network conditions. Simulation data confirm the effectiveness of the method in such environments. Even when applied to larger-scale maritime networks, where node density remains relatively limited, the method maintains its applicability and efficiency.
Comment 11:While the hybrid passive-active + Q-learning scheme is innovative in theory, its real-world viability is limited by energy, computation, and protocol stack constraints inherent to opportunistic and delay-tolerant networks. This work will benefit from reduction in algorithmic complexity; Incorporate social/temporal utility into discovery; Integrate with realistic MAC protocols; Extend simulations to diverse environments.
Response 11:I sincerely appreciate the reviewer’s highly valuable comments and suggestions on this paper. Since the proposed neighbor discovery method is specifically designed for maritime internet networks—a core focus of my research—it is tailored to opportunistic networking characteristics commonly found in marine environments. As a result, its applicability in broader real-world scenarios may be somewhat limited.It is true that the initial phase of neighbor discovery in this method requires relatively high energy consumption to complete the initialization process. This setup, however, enables subsequent power optimization through Q-learning. Incorporating additional influencing factors in future work could help extend the approach’s effectiveness to a wider range of network scenarios.Once again, I thank the reviewer for their insightful recommendations, which have greatly helped improve this study.
Comment 12:The research gaps is vague, and no clear contribution summary was set out. Authors should confirm how their proposal is different from these works
Response 12:Thank you for pointing this out. I will carefully revise my manuscript to clearly highlight the contributions of this work in the abstract, introduction, and discussion sections. Specifically, I will emphasize the improvements and limitations of the proposed method. In addition, I will thoroughly review the literature you recommended to further strengthen the arguments and context of the study.I truly appreciate the time and effort the reviewers have dedicated to providing insightful comments on my paper.
Reviewer 2 Report
Comments and Suggestions for Authors
Please see the attachment.

Author Response
Comments 1:The introduction is too brief. It is recommended to expand it by including a more detailed discussion on concrete applications of opportunistic networks to better contextualize the study and enhance its relevance.
Response 1:Thank you for your valuable suggestions. Since my research focuses specifically on maritime internet networks, the proposed method is designed for and limited to maritime opportunistic networking scenarios. I recognize that my previous assumption of its general applicability to common network environments was inappropriate, and I have revised the manuscript accordingly. I sincerely apologize for this oversight.I greatly appreciate your feedback, which has significantly improved the relevance and precision of this study.
Comments 2:The related work section should be enriched with additional references to better position the article’s originality and strengthen its credibility by highlighting specific advances in opportunistic networks.
Response 2:Thank you for your suggestion. I acknowledge that much of the existing literature on neighbor discovery in opportunistic networks is somewhat dated, and recent studies in this area remain relatively limited. Following your advice, I will focus on improving the work by incorporating more up-to-date references and perspectives. I will also continue to actively explore recent publications to deepen my understanding and contribute more effectively to this field.
Comments 3:A comprehensive, fully labeled system diagram is essential to facilitate the understanding of the methodology and fully grasp the authors’ ideas. Although several partial diagrams are provided, the absence of a synthetic overall scenario diagram reduces clarity.
Response 3:Thank you for your suggestion. Due to possible version limitations, I have not yet found a method to export high-resolution scenario images in Exata, and I apologize for this issue. I will continue to explore alternative approaches for generating high-quality figures. I appreciate your understanding and your valuable advice.
Comments 4:The quality of the figures needs improvement. From Figure 5 onward, some images, especially those showing simulation results, appear blurry, affecting their readability and data interpretation.
Response 4:Thank you for your suggestion. I have increased the resolution of the figures to ensure the results are presented more clearly.
Comments 5:Subsection 5.3, dedicated to the discussion of simulation results, is too brief. It should be expanded to provide a more detailed and critical analysis of the presented results.
Response 5:Thank you for your suggestion. I have further elaborated on both the advantages and the limitations of the proposed method in the Discussion and Conclusion sections. Your feedback has been invaluable in improving the depth and balance of the manuscript.
Comments 6:It is recommended to add a comparative table of results from related works. This would better situate the advances made by this article by clearly highlighting its specific contributions and originality.
Response 6:Thank you for your feedback. I have added a new section in the introduction highlighting the innovative aspects and contributions of the proposed method. The experimental results, presented in tabular form, provide a clear and intuitive demonstration of the improvements achieved.Since this approach builds upon an existing method and there are relatively few reference studies specifically focused on opportunistic networks, further development remains ongoing. I will continue to study relevant literature and explore this field to seek new insights in future work. I truly appreciate your valuable suggestions.
Comments 7:The authors should discuss the limitations of their approach and potential ways to address them. This would offer a more comprehensive view of the work and open perspectives for improvement and future research.
Response 7:Thank you for your suggestion. I have elaborated on both the strengths and the limitations of the proposed method in the Discussion section. In the Conclusion, I have also outlined potential future research directions and further developments. I greatly appreciate your insightful comments.
Comments 8:Finally, it would be beneficial to include quantitative results in the conclusion to strengthen the impact and clarity of the study’s contributions.
Response 8:Thank you for your suggestion. I will revise the relevant sections of the article accordingly to address this point in the conclusion.
Reviewer 3 Report
Comments and Suggestions for Authors
Dear Authors
Your paper is near identical in content to reference [13] (https://ieeexplore.ieee.org/abstract/document/11069539), which you have authored and therefore I cannot recommend it for publication.
According to MDPI R rules
"Manuscripts submitted to MDPI journals should meet the highest standards of publication ethics:
- Manuscripts should only report results that have not been submitted or published before, even in part.
Author Response
I am sorry to hear that the previous version of the manuscript did not fully meet your expectations. Based on feedback from all reviewers, the paper has undergone significant revisions to better address key concerns.
First, the proposed neighbor discovery method is now explicitly contextualized within maritime opportunistic networks, rather than being presented as a general solution. Realistic maritime channel models have been incorporated to more accurately simulate the oceanic communication environment, improving both relevance and reproducibility.Second, a neighbor table update mechanism has been added to limit memory overhead caused by excessive neighbor information. This enhancement helps maintain scalability under dynamic network conditions.Third, comprehensive experimental analyses have been introduced to evaluate how different active probing power levels affect critical performance metrics, including throughput, energy consumption, latency, and packet loss rate. These results provide deeper insight into the trade-offs involved in power-aware neighbor discovery.Finally, the conclusion now includes a balanced discussion of the advantages and limitations of the method, as well as promising directions for future improvement.Thank you very much for your time and effort in reviewing this manuscript. Your feedback has been invaluable in shaping these revisions.
Round 2
Reviewer 1 Report
Comments and Suggestions for Authors
It looks like the authors have addressed my concerns
Reviewer 2 Report
Comments and Suggestions for Authors
The authors have addressed the majority of the comments and have made the necessary efforts to implement them.
Reviewer 3 Report
Comments and Suggestions for Authors
Dear Authors.
You have satisfactorily addressed my concerns. I wish you the best of luck with your future work.